# Potential Common Mechanisms of Cytotoxicity Induced by Amide Herbicides via TRPA1 Channel Activation

**DOI:** 10.3390/ijerph19137985

**Published:** 2022-06-29

**Authors:** Xiaoning Wang, Yangyang Sun, Qian Wang, Fengying Liu, Weijie Yang, Xin Sui, Jun Yang, Minmin Zhang, Shuai Wang, Zhenyu Xiao, Yuan Luo, Yongan Wang, Tong Zhu

**Affiliations:** 1Institute of Process Equipment and Environmental Engineering, School of Mechanical Engineering and Automation, Northeastern University, Shenyang 110819, China; wangxn5593@163.com; 2State Key Laboratory of Toxicology and Medical Countermeasures, Beijing Institute of Pharmacology and Toxicology, Beijing 100850, China; sunyangsongyang@126.com (Y.S.); 307396444@163.com (Q.W.); lly610_yy@163.com (F.L.); yangwj_1997@163.com (W.Y.); sx_226@163.com (X.S.); ajaway@126.com (J.Y.); ozhangmm@163.com (M.Z.); ws20095072016@163.com (S.W.); zhenyuxiao2021@163.com (Z.X.); yonganw@126.com (Y.W.)

**Keywords:** amide herbicides, TRPA1 channel, cytotoxicity, computational simulation, common mechanisms

## Abstract

The “Multi-Threat Medical Countermeasure (MTMC)” strategy was proposed to develop a single drug with therapeutic efficacy against multiple pathologies or broad-spectrum protection against various toxins with common biochemical signals, molecular mediators, or cellular processes. This study demonstrated that cytotoxicity, expression of transient receptor potential cation channel subfamily A member 1 (TRPA1) mRNA, and intracellular calcium influx were increased in A549 cells exposed to amide herbicides (AHs), in which the order of cytotoxicity was metolachlor > acetochlor > propisochlor > alachlor > butachlor > propanil > pretilachlor, based on IC_50_ values of 430, 524, 564, 565, 619, 831, and 2333 μM, respectively. Inhibition/knockout of TRPA1 efficiently protected against cytotoxicity, decreased TRPA1 mRNA expression, and reduced calcium influx. The results suggested that the TRPA1 channel could be a key common target for AHs poisoning. The order of TRPA1 affinity for AHs was propanil > pretilachlor > metolachlor > (propiso/ala/aceto/butachlor), based on K_D_ values of 16.2, 309, and 364 μM, respectively. The common molecular mechanisms of TRPA1-AHs interactions were clarified, including toxicity-effector groups (benzene ring, nitrogen/oxygen-containing functional groups, halogen) and residues involved in interactions (Lys787, Leu982). This work provides valuable information for the development of TRPA1 as a promising therapeutic target for broad-spectrum antitoxins.

## 1. Introduction

In recent decades, agricultural pesticides have been investigated by environmental or toxicology researchers due to their wide use and accumulation in ecosystems, evident toxic effects on human health, and environmental safety concerns [1,2,3]. Consequently, medical prevention strategies against multifarious chemical poisonings are being explored. The traditional model of drug discovery provides a specific medical countermeasure against a single symptom or pathology of a chemical toxin but is not suitable against multiple toxins [4,5]. Alternatively, the “Multi-Threat Medical Countermeasure (MTMC)” strategy supports the possibility of developing a single countermeasure drug with prophylactic or therapeutic efficacy against signaling pathways or inflammation pathologies caused by multiple toxins [4,6]. This study aimed to investigate the common interaction mechanisms within a family of toxic compounds, laying the groundwork for future research based on MTMC.

Various inflammatory reactions are caused by toxic compounds via cytomembrane or organelle receptors [7,8,9]. Transient receptor potential ion channels (TRPs), especially TRP subfamily A member 1 (TRPA1), are membrane proteins with high permeability to calcium that are considered as “chemical switches”, as the targets of many environmental aromatic compounds (e.g., cinnamaldehyde, allicin, formaldehyde, acrolein, allyl isothiocyanate, etc.) and proinflammatory agents that induce inflammatory reactions [10,11,12,13]. Specifically, TRPA1 ankyrin-like repeat domain (TRPA1 ARD), within the N-terminal domain of human TRPA1 (hTRPA1), interacts with electrophilic aromatic molecules, which leads to intracellular calcium influx and triggers a series of intracellular cascade reactions or apoptosis [14,15,16,17,18]. Thus, a common mechanism underlying the effects of toxins on biological systems may be determined by elucidating the interactions between TRPA1 and aromatic compound-containing toxins.

Amide herbicides (AHs), such as propanil, propisochlor, metolachlor, alachlor, acetochlor, pretilachlor, and butachlor, are members of the chloroacetanilide family and contain benzene rings. AHs are used globally in agricultural environments and contribute to endocrine disruption and inflammatory reactions [8,19,20,21,22,23]. Although many patients are hospitalized due to improper use of AHs, specific therapeutic drugs are currently lacking [24,25,26,27,28,29]. Therefore, the aim of this study was to explore the common biochemical pathways of AHs poisoning on an effort to improve treatment options.

The mainstream mode of detoxification following compound poisoning is to first clarify the toxin and target using laboratory technology or computer simulation [30,31]. Evaluating toxicity to cellular materials or proteins using the above techniques could help determine common mechanisms by exploring interactions between compounds and targets [32,33,34,35]. Therefore, this study provides valuable information regarding a promising therapeutic target for the development of specific broad-spectrum antitoxins against AHs poisoning or other compounds designed to inhibit target bioactivity.

## 2. Materials and Methods

### 2.1. Chemicals

Human lung type II epithelial A549 cells were purchased from the Cell Resource Center, Institute of Basic Medicine, Chinese Academy of Medical Sciences (Beijing, China). Seven AHs, including propanil, propisochlor, metolachlor, alachlor, acetochlor, pretilachlor, and butachlor were purchased from Shanghai Aladdin Biochemical Technology Co., Ltd. (Shanghai, China). The TRP channel inhibitor HC-030031 was purchased from Selleck Biotechnology Co., Ltd. (Houston, TX, USA). Cell Counting Kit-8 (CCK-8) and Cytotoxicity LDH Assay Kit-WST were purchased from Dojindo Laboratories (Kumamoto, Japan). TRIzol Reagent and Fluo-4 Direct calcium assay kits were purchased from Invitrogen Trading Co., Ltd. (Shanghai, China). PrimeScript RT reagent kit with gDNA Eraser and TB Green Premix Ex Taq II were purchased from Takara Bio Inc. (Kusatsu, Japan). hTRPA1 ARD protein was purchased from Immuno Clone Biosciences Co., Ltd. (Huntington Station, NY, USA).

### 2.2. Construction of TRPA1 Gene Knockout (TRPA1-KO) A549 Cells

The *hTRPA1* gene sequence (Gene ID: 8989) was obtained from the National Center for Biotechnology Information (NCBI) website (https://www.ncbi.nlm.nih.gov/gene/8989 accessed on 26 May 2022). Three gRNA sequences, namely, T1: 5′-GAATGGATTATCTACACGAC-302B9, T16: 5′-GTAATTGGACATTTATTGCC-3′, and T20: 5′-TTTTATATTGACAACGAGAA-3′, were designed using the CRISPR online design tool (http://crispr.mit.edu/ accessed on 26 May 2022). Transfection of cells with gRNA-Cas9 was validated in test cells. Cell pool examination was performed using Sanger sequencing. Sequencing and tracer analysis were performed using the online tool TIDE (https://tide-calculator.nki.nl/ accessed on 26 May 2022) to investigate the cleavage efficiency of T1, T16, and T20. Cells from the cell pool were diluted in 96-well plates. Clones were observed under a microscope and wells containing single cell clones were labeled. Single cell clones were transferred to 24-well plates for expansion. Validation of knockout (KO) clones was performed via sequencing to confirm insertion and deletion on the genomes of KO clone candidates. Furthermore, mycoplasmas were detected for cloned cells with high KO using the Mycoplasma PCR Detection Kit.

### 2.3. Cell Cultures and Treatment

Wild-type (WT) and *TRPA1*-KO A549 cells were cultured in McCoy’s 5A Media and F-12K Nutrient Mixture containing 10% fetal bovine serum, respectively. WT/*TRPA1*-KO A549 cells (5 × 10^4^ cells/mL) were seeded in 96-well (100 μL/well) and 6-well (2 mL/well) plates and incubated at 37 °C with 5% CO_2_ in a humidified incubator for 24 h. WT/*TRPA1*-KO A549 cells were treated with AHs gradient concentrations (0, 150, 300, 600, 1200 μM) prepared in methanol (1.2 M) for 24 h. Additionally, WT/*TRPA1*-KO A549 cells were pre-treated with TRPA1 special inhibitor HC-030031 (100 μM) for 0.5 h and then treated with AHs gradient concentrations for 24 h.

### 2.4. Cell Viability Assay

To conduct the cell viability assay, the growth medium was removed from WT/*TRPA1*-KO A549 cells cultured in a 96-well plate. CCK-8 reagent was mixed with complete medium (1:9 volume ratio), and 100 µL of the CCK-8 reagent mixture was added to each well. The plate was incubated for 4 h and the absorbance was measured at 450 nm using a microplate reader (BNR05740; Molecular Devices, Sunnyvale, CA, USA). The average absorbance from each sextuplicate set of wells was calculated and the background control value was subtracted from each absorbance value. The percentage cell viability was determined using the following equation:cell viability (%) = [(As − Ab)/(Ac − Ab)] × 100%
where As is the absorbance of the experimental well (containing medium, CCK-8 reagent, cells, and compound); Ac is the absorbance of the control well (containing medium, CCK-8 reagent, and cells); and Ab is the absorbance of the blank well (containing medium and CCK-8 reagent).

### 2.5. Cytotoxicity Assay

Using the Cytotoxicity LDH Assay Kit, lysis buffer (10 μL) was added to the high control well of a 96-well plate containing WT/*TRPA1*-KO A549 cells, which was then incubated for 0.5 h. Working solution was prepared by adding 5 mL assay buffer to the dye mixture vial, of which 100 μL was added to each well. The plate was protected from light and incubated at room temperature for 0.5 h. Subsequently, 50 μL stop solution was added to each well and the absorbance was measured immediately at 490 nm using a microplate reader (BNR05740, Molecular Devices). The average absorbance was calculated from each sextuplicate set of wells and the background control value was subtracted from each absorbance value. The percentage cytotoxicity was determined using the following equation:cytotoxicity (%) = [(Aa − Ab1)/(Ab2 − Ab1)] × 100%
where Aa is the absorbance of the experimental well (containing medium, cells, and compound); Ab2 is the absorbance of the high control well (containing medium, cells, and lysis buffer); and Ab1 is the absorbance of the low control well (containing medium and cells).

### 2.6. Real-Time Quantitative PCR (RT-qPCR)

TRIzol Reagent (1 mL) was directly added to WT/*TRPA1*-KO A549 cells in 6-well plates and total RNA was extracted according to the manufacturer’s instructions. Total RNA was used for cDNA reverse transcription, which was performed according to the PrimeScript RT protocol with the following reaction conditions: 42 °C for 2 min, 4 °C for ∞ (removal of genomic DNA), 37 °C for 15 min, 85 °C for 5 s, and 4 °C for ∞ (reverse transcriptional reaction). RT-qPCR was performed according to the TB Green Premix Ex Taq II protocol. Primer sequences and product lengths were as follows: hTRPA1 (107 bp) forward primer 5′-AGTATATTTGGGTATTGCAAAGAAGC-3′, reverse primer 5′-ATGCCCGTCGTGTAGATAATCC-3′; hβ-actin (184 bp) forward primer 5′-AGAGCTACGAGCTGCCTGAC-3′, reverse primer 5′-AGCACTGTGTTGGCGTACAG-3′. RT-qPCR was performed using the CFX96 Real-Time System (Bio-Rad Laboratories, Hercules, CA, USA) using the following reaction conditions: 95 °C for 30 s, 40 cycles of 95 °C for 5 s, 60 °C for 30 s, and 72 °C for 30 s, followed by a melting curve from 65 to 95 °C in 0.5 °C/5 s increments. Calculations were performed using the CFX Manager^TM^ Version 1.0 software (Bio-Rad Laboratories Ltd., Hercules, CA, USA) and are represented as fold change in expression [2^ΔΔC(t)^] on a linear scale.

### 2.7. Cellular Calcium Imaging

Working solution was prepared according to the instructions of the Fluo-4 Direct calcium assay kit and directly added to WT/*TRPA1*-KO A549 cells in 96-well plates, which were incubated for 0.5 h and kept away from light after removal from the incubator. Single channel fluorescein isothiocyanate (FITC) levels were measured using a confocal high-content imaging system (IXM-C S150069, Molecular Devices, San Jose, CA, USA).

### 2.8. Two-Stage Mass Spectrometry (MS/MS) Analysis

TRPA1 recombinant protein were separated by SDS-PAGE, stained with CBB R250, and TRPA1 protein bands were precisely excised by a sharp blade on the clean glass plate. Peptides (2 μL) were analyzed at high resolution using a NanoLC-Orbitrap Elite MS/MS system (Thermo Fisher Scientific, Waltham, MA, USA) with an automatic sampler for data acquisition. The system was operated in positive ion spray ionization mode, with an injection speed of 0.3 µL/min, capillary voltage of 2.2 kV, mass-to-charge ratio range of 300–1800 *m*/*z*, and 35% collision energy. Data processing was performed using Proteome Discovery 2.4 software. Qualitative assessment of proteins and mass-to-charge ratios (*m*/*z*) of fragment peaks were performed by comparing differences in MS/MS spectra.

### 2.9. Localized Surface Plasmon Resonance (LSPR) Analysis

The binding capacity of hTRPA1 ARD recombinant protein and AHs was determined by LSPR according to the OpenSPR (Nicoya Life Sciences, Kitchener, ON, Canada) operating procedure using the COOH chip. After the baseline signal was obtained, the sample was washed with 200 µL 80% isopropyl alcohol for 10 s, and 200 µL ethylcarbodiimide/N-hydroxysuccinimide solution was added. The sample loop was rinsed with phosphate-buffered saline (PBS). hTRPA1 (5 µL) was diluted with PBS, incubated for 4 min, and 200 µL blocking solution was added. The sample loop was observed at baseline for 5 min. The following analytes were used: propanil (0, 1.25, 2.5, 5, 10 μM), propisochlor (0, 10, 80, 100, 1000 μM), metolachlor (0, 25, 50, 100, 200 μM), alachlor (0, 250, 1000 μM), acetochlor (0, 100, 500, 1000, 2000 μM), pretilachlor (0, 500, 1000, 2000 μM), and butachlor (0, 10, 100, 1000, 2000 μM). Contact and dissociation time of analytes and protein were 240 s and 160 s, respectively. TraceDrawer software was applied to determine the association constant (k_a_), dissociation constant (k_d_), and affinity constant (K_D_) using a 1:1 model.

### 2.10. Molecular Docking

The macromolecular structure of hTRPA1 used for molecular docking simulations was obtained from the Protein Data Bank (PDB ID 6PQO, https://www.rcsb.org accessed on 26 May 2022) and undesired structures were managed using the SEQ module of Molecular Operating Environment (MOE) version 20.09 software [36]. The structures of AHs (propanil, propisochlor, metolachlor, alachlor, acetochlor, pretilachlor, and butachlor) were derived from the PubChem database (https://pubchem.ncbi.nlm.nih.gov/ accessed on 26 May 2022) in SDF format. Small molecules were minimized and saved. Macromolecule docking pockets were determined using the Site Finder module. Molecular docking simulations were performed using the Docking module and all docking processes were completed under an Amber 10: EHT force field, using an R-field dominant solvent model with pH 7.0 and temperature 300 K. The GBVI/WSA dG scoring function was used to score 30 structures with London dG scores for flexible docking. Macromolecules and small molecules were further screened for better interaction performance.

### 2.11. Statistical Analysis

The experimental data are expressed as ± standard error of mean (SEM). GraphPad Prism version 6.0 software (GraphPad Software, La Jolla, CA, USA) was used for data analysis. One-way analysis of variance for multiple comparisons was used to analyze differences between groups, and *p* < 0.05 was considered statistically significant.

## 3. Results

### 3.1. Construction of the TRPA1-KO A549 Cell Line

The *TRPA1*-KO A549 cell line was constructed using CRISPR/Cas9 genome editing. The cleavage efficiency of gRNAs was 24.4% (T1), 0% (T16), and 0% (T20). Based on these results, T1 gRNA was selected for subsequent tests. Following KO of the *TRPA1* gene by T1 gRNA, double allele KO of the *TRPA1* gene in cell clones was confirmed by DNA sequencing. The results of genotyping for a single cell were clone T1-2 (A549 *TRPA1* 0/0), clone T1-20 (A549 *TRPA1* +1/+1), and clone T1-53 (A549 *TRPA1* −25/−25) (Appendix A). Mycoplasmas were not detected for *TRPA1-*KO positive clone T1-20 (A549 *TRPA1* +1/+1), T1-53 (A549 *TRPA1* −25/−25), and *TRPA1-*KO negative clone T1-2 (A549 *TRPA1* 0/0) (Appendix A). Finally, T1-53 (A549 *TRPA1* −25/−25) cells were selected for subsequent experiments.

### 3.2. AHs Cytotoxicity towards A549 Cells

#### 3.2.1. AHs Treatment Decreases the Viability of A549 Cells

To determine the cytotoxic effects of AHs on A549 cells, cell viability was evaluated using CCK-8 reagent after treatment with different concentrations of AHs for 24 h. Compared with treatment with 0 μM AHs, the viability of A549 cells treated with propanil (150 μM, *p* < 0.001; 300 μM, *p* < 0.0001; 600 μM, *p* < 0.0001; 1200 μM, *p* < 0.0001) (Appendix A), propisochlor (150 μM, *p* < 0.01; 300 μM, *p* < 0.0001; 600 μM, *p* < 0.0001; 1200 μM, *p* < 0.0001) (Appendix A), metolachlor (all *p* < 0.0001) (Appendix A), alachlor (150 μM, *p* < 0.001; 300 μM, *p* < 0.001; 600 μM, *p* < 0.0001; 1200 μM, *p* < 0.0001) (Appendix A), acetochlor (all *p* < 0.0001) (Appendix A), pretilachlor (150 μM, *p* < 0.001; 300 μM, *p* < 0.001; 600 μM, *p* < 0.001; 1200 μM, *p* < 0.0001) (Appendix A), and butachlor (150 μM, *p* < 0.01; 300 μM, *p* < 0.0001; 600 μM, *p* < 0.0001; 1200 μM, *p* < 0.0001) (Appendix A) was significantly decreased.

#### 3.2.2. AHs Are Cytotoxic to A549 Cells

The cytotoxic effects of AHs on A549 cells were further evaluated using the LDH assay after treatment with different concentrations of AHs for 24 h. Compared with treatment with 0 μM AHs, A549 cells treated with propanil (all *p* < 0.0001) (Appendix A), propisochlor (all *p* < 0.0001) (Appendix A), metolachlor (all *p* < 0.0001) (Appendix A), alachlor (all *p* < 0.0001) (Appendix A), acetochlor (all *p* < 0.0001) (Appendix A), pretilachlor (all *p* < 0.0001) (Appendix A), and butachlor (150 μM, *p* < 0.01; 300 μM, *p* < 0.0001; 600 μM, *p* < 0.0001; 1200 μM, *p* < 0.0001) (Appendix A) exhibited significantly increased cell damage. These findings indicated that seven AHs demonstrated obvious cytotoxic effects on A549 cells.

#### 3.2.3. AHs Treatment Increased TRPA1 mRNA Expression in A549 Cells

Although preliminary research revealed the cytotoxicity of AHs towards A549 cells, the target of AHs poisoning remained unclear. Activated by allyl isothiocyanate (AITC), delta-9-tetrahydrocannabinol, acrolein, and other compounds with electrophilic organic characteristics, previous studies have identified that TRPA1 ion channels act as “switches” for multiple types of chemical poisoning, resulting in abnormal physiological and biochemical reactions [7,9]. Therefore, the current study explored whether TRPA1 is a common target for several different AHs. Compared with that in 0 μM AHs-treated cells, the expression of TRPA1 mRNA in A549 cells treated with propanil (300 μM, *p* < 0.05; 600 μM, *p* < 0.01) (Appendix A), propisochlor (300 μM, *p* < 0.01; 600 μM, *p* < 0.01) (Appendix A), metolachlor (300 μM, *p* < 0.01; 600 μM, *p* < 0.0001) (Appendix A), alachlor (300 μM, *p* < 0.01; 600 μM, *p* < 0.01) (Appendix A), acetochlor (600 μM, *p* < 0.01) (Appendix A), pretilachlor (600 μM, *p* < 0.05) (Appendix A), and butachlor (150 μM, *p* < 0.05; 300 μM, *p* < 0.05; 600 μM, *p* < 0.001) (Appendix A) was significantly increased. Thus, treatment with seven AHs significantly increased TRPA1 mRNA expression in A549 cells.

### 3.3. Exploring the Target of AHs in A549 Cells

#### 3.3.1. AHs Treatment Increased TRPA1 mRNA Expression in A549 Cells, Which Is Reversed by HC-030031 Intervention or TRPA1-KO

Changes in AHs-induced cytotoxicity were measured after blocking or knocking out TRPA1 to determine if TRPA1 is a key target involved in AHs poisoning. Compared with AHs treatment alone (150, 300, 600 μM), A549 cells pre-treated with HC-030031 (50 μM, TRPA1-specific inhibitor) exhibited significantly reduced TRPA1 mRNA expression after treatment with propanil (150 μM, *p* < 0.01) (Figure 1A), metolachlor (150 μM, *p* < 0.05) (Figure 1C), alachlor (600 μM, *p* < 0.05) (Figure 1D), acetochlor (150 μM, *p* < 0.01; 600 μM, *p* < 0.05) (Figure 1E), and butachlor (600 μM, *p* < 0.01) (Figure 1G). Pre-treatment with HC-030031 (50 μM) also decreased TRPA1 mRNA expression in A549 cells treated with propisochlor (Figure 1B) and pretilachlor (Figure 1F), albeit not significantly. Furthermore, expression of TRPA1 mRNA was not detected in *TRPA1*-KO A549 cells, even in those treated with AHs (Figure 1A–G). These results indicated that HC-030031 intervention or *TRPA1*-KO could decrease TRPA1 mRNA expression in AHs-treated A549 cells.

#### 3.3.2. Viability of AHs-Treated A549 Cells Is Increased with HC-030031 Intervention or TRPA1-KO

Compared with AHs treatment alone (150, 300, 600 μM), A549 cells pre-treated with HC-030031 (50 μM) exhibited significantly increased viability when cells were treated with propanil (150 μM, *p* < 0.05; 300 μM, *p* < 0.05; 600 μM, *p* < 0.01) (Figure 2A), metolachlor (150 μM, *p* < 0.01; 600 μM, *p* < 0.01) (Figure 2C), alachlor (300 μM, *p* < 0.0001; 600 μM, *p* < 0.0001) (Figure 2D), acetochlor (150 μM, *p* < 0.001; 300 μM, *p* < 0.01; 600 μM, *p* < 0.001) (Figure 2E), pretilachlor (600 μM, *p* < 0.05) (Figure 2F), and butachlor (300 μM, *p* < 0.01) (Figure 2G). Moreover, the viability of A549 cells pre-treated with HC-030031 (50 μM) was increased when cells were treated with propisochlor (Figure 2B), albeit not significantly. Similarly, KO of *TRPA1* significantly increased the viability of A549 cells exposed to propanil (300 μM, *p* < 0.05; 600 μM, *p* < 0.01) (Figure 2A), propisochlor (150 μM, *p* < 0.05; 600 μM, *p* < 0.01) (Figure 2B), metolachlor (150 μM, *p* < 0.001; 600 μM, *p* < 0.01) (Figure 2C), alachlor (150 μM, *p* < 0.01; 300 μM, *p* < 0.01; 600 μM, *p* < 0.05) (Figure 2D), acetochlor (150 μM, *p* < 0.001; 300 μM, *p* < 0.0001; 600 μM, *p* < 0.05) (Figure 2E), pretilachlor (150 μM, *p* < 0.05) (Figure 2F), and butachlor (150 μM, *p* < 0.01; 300 μM, *p* < 0.001; 600 μM, *p* < 0.01) (Figure 2G).

The IC_50_ values of AHs for WT A549 cells (from strongest to weakest) were in the following order: metolachlor (430 µM) > acetochlor (524 µM) > propisochlor (564 µM) > alachlor (565 µM) > butachlor (619 µM) > propanil (831 µM) > pretilachlor (2333 µM) (Table 1). AHs had a pronounced cytotoxic effect on A549 cells, among which metolachlor demonstrated the strongest cytotoxicity. Moreover, the IC_50_ values of propanil, propisochlor, metolachlor, alachlor, acetochlor, pretilachlor, and butachlor for A549 cells with HC-030031 (50 μM) intervention were 2290, 1690, 620, 731, 647, 4456, and 1757 μM, respectively, and 2558, 1828, 562, 659, 696, 3672, and 767 μM for *TRPA1*-KO A549 cells, respectively. The IC_50_ values of AHs with HC-030031 intervention or *TRPA1*-KO for A549 cells were increased compared with those of WT A549 cells, and cytotoxicity was decreased. The results indicated that TRPA1 blocking through chemical intervention or *TRPA1*-KO could increase the viability of A549 cells treated with AHs, confirming that cytotoxicity induced by AHs occurred via the TRPA1 channel.

#### 3.3.3. Calcium Influx in AHs-Treated A549 Cells Is Decreased with HC-030031 Intervention or TRPA1-KO

Activation of TRPA1 by chemical compounds induces cationic influx through open channel pores that are highly permeable to calcium ions. Cytotoxicity mediated by TRPA1 appears to depend on an increase in intracellular calcium levels and is related to apoptosis-like, calcium-dependent mechanisms [13]. Thus, intracellular calcium influx was measured by cellular calcium imaging in AHs-treated A549 cells. Compared with that in 0 μM AHs-treated cells, calcium influx was significantly increased in A549 cells treated with propanil (150 μM, *p* < 0.01; 300 μM, *p* < 0.001; 600 μM, *p* < 0.001; 1200 μM, *p* < 0.0001) (Figure 3A), propisochlor (150 μM, *p* < 0.01; 300 μM, *p* < 0.0001; 600 μM, *p* < 0.0001; 1200 μM, *p* < 0.0001) (Figure 3B), metolachlor (150 μM, *p* < 0.001; 300 μM, *p* < 0.0001; 600 μM, *p* < 0.0001; 1200 μM, *p* < 0.0001) (Figure 3C), alachlor (all *p* < 0.0001) (Figure 3D), acetochlor (all *p* < 0.0001) (Figure 3E), pretilachlor (all *p* < 0.0001) (Figure 3F), and butachlor (150 μM, *p* < 0.001; 300 μM, *p* < 0.0001; 600 μM, *p* < 0.0001; 1200 μM, *p* < 0.0001) (Figure 3G).

Furthermore, compared with AHs treatment alone (150, 300, 600, 1200 μM), pre-treatment with HC-030031 (50 μM) significantly decreased calcium influx in A549 cells treated with propanil (300 μM, *p* < 0.05; 600 μM, *p* < 0.01) (Figure 3A), propisochlor (150 μM, *p* < 0.01; 300 μM, *p* < 0.0001; 600 μM, *p* < 0.0001; 1200 μM, *p* < 0.01) (Figure 3B), metolachlor (150 μM, *p* < 0.01; 300 μM, *p* < 0.05; 600 μM, *p* < 0.001; 1200 μM, *p* < 0.0001) (Figure 3C), alachlor (150 μM, *p* < 0.01; 300 μM, *p* < 0.0001; 600 μM, *p* < 0.0001) (Figure 3D), acetochlor (all *p* < 0.0001) (Figure 3E), pretilachlor (150 μM, *p* < 0.05; 300 μM, *p* < 0.001; 600 μM, *p* < 0.0001) (Figure 3F), and butachlor (300 μM, *p* < 0.0001; 600 μM, *p* < 0.0001; 1200 μM, *p* < 0.05) (Figure 3G). Similarly, KO of *TRPA1* significantly decreased calcium influx in A549 cells treated with propanil (150 μM, *p* < 0.001; 600 μM, *p* < 0.05; 1200 μM, *p* < 0.0001) (Figure 3A), propisochlor (150 μM, *p* < 0.0001; 300 μM, *p* < 0.001; 600 μM, *p* < 0.0001; 1200 μM, *p* < 0.001) (Figure 3B), metolachlor (150 μM, *p* < 0.0001; 300 μM, *p* < 0.001; 600 μM, *p* < 0.0001; 1200 μM, *p* < 0.01) (Figure 3C), alachlor (150 μM, *p* < 0.001; 300 μM, *p* < 0.0001; 600 μM, *p* < 0.01; 1200 μM, *p* < 0.0001) (Figure 3D), acetochlor (all *p* < 0.0001) (Figure 3E), pretilachlor (150 μM, *p* < 0.0001; 300 μM, *p* < 0.0001; 600 μM, *p* < 0.001; 1200 μM, *p* < 0.0001) (Figure 3F), and butachlor (150 μM, *p* < 0.0001; 300 μM, *p* < 0.0001; 600 μM, *p* < 0.0001; 1200 μM, *p* < 0.01) (Figure 3G). To summarize, TRPA1 was an important target mediating calcium influx in A549 cells treated with AHs, while TRPA1 blocking through chemical intervention or *TRPA1*-KO could inhibit calcium influx in A549 cells.

### 3.4. Molecular Mechanism of TRPA1-AHs Interaction

#### 3.4.1. MS/MS Results

The amino acid sequence of hTRPA1 ARD protein was confirmed by MS/MS. SDS-PAGE analysis indicated that the molecular weight of hTRPA1 ARD was 37.6 kDa (Appendix A). The secondary structure determined by collision-induced dissociation (CID) and mass-to-electricity ratios (*m*/*z*) determined by MS/MS after enzymology (Appendix A) indicated that the molecular weight and amino acid sequence of the purchased hTRPA1 ARD was consistent with the corresponding sequence on the NCBI website (Appendix A).

#### 3.4.2. Affinity of hTRPA1 for AHs

LSPR analysis indicated that k_a_, k_d_, and K_D_ values of hTRPA1-propanil were 2390 M^−1^·s^−1^, 0.0388 s^−1^, and 16.2 μM, respectively (Figure 4A), whereas those for hTRPA1-metolachlor were 112 M^−1^·s^−1^, 0.0408 s^−1^, and 364 μM, respectively (Figure 4C); and those for hTRPA1-pretilachlor were 14.3 M^−1^·s^−1^, 0.00443 s^−1^, and 309 μM, respectively (Figure 4F). Obvious binding was not detected between hTRPA1 and propisochlor, alachlor, acetochlor, or butachlor at the above-mentioned concentrations (Figure 4B,D,E,G). The order of k_a_ values for AHs-hTRPA1 was propanil > metolachlor > pretilachlor. The order of k_d_ values for AHs-hTRPA1 was pretilachlor > propanil > metolachlor. The order of K_D_ values for AHs-hTRPA1 was propanil > pretilachlor > metolachlor. In summary, propanil, pretilachlor, and metolachlor showed stronger binding to hTRPA1, whereas no obvious affinity was observed between hTRPA1 and propisochlor, alachlor, acetochlor, or butachlor.

#### 3.4.3. Common Mechanism of hTRPA1-AHs Interactions

Common interaction mechanisms between hTRPA1 and AHs were clarified in detail using computational docking simulations. The AHs contained hydrophobic (benzene ring, hydrocarbon chain) and active (–Cl, –C(=O)– or –O–, –N– or –NH–) groups (Appendix A). The main structural differences among AHs included side chain groups, hydrocarbon chains, and halogenic content. The following residues were involved in interactions between hTRPA1 and AHs: Asn722, Lys787, Gln791, Ser804, Met978, and Leu982 for hTRPA1-propanil; Lys787, Gln791, Ile803, Met844, Arg852, Lys868, Leu982, and Lys989 for hTRPA1-propisochlor; Lys787, Gln791, Tyr840, Met844, Arg852, Ile860, Glu864, Lys868, Lys974, Leu982, and Lys989 for hTRPA1-metolachlor; Asn722, Ser725, Lys787, Gln791, Glu864, Lys868, and Leu982 for hTRPA1-alachlor; Asn722, Lys787, Gln791, Met844, Arg852, Lys868, and Leu982 for hTRPA1-acetochlor; Asn722, Lys787, Ile803, Ser804, Met844, Glu864, Lys868, and Leu982 for hTRPA1-pretilachlor; and Asn722, Lys787, Gln791, Asp802, Ile803, Ser804, Glu808, Glu864, Lys868, and Leu982 for hTRPA1-butachlor. The key toxicity-effector groups of each AHs were identified as follows: propanil possessed a benzene ring, –C(=O)–, –NH–, acceptor (Acc), aromatic (Aro), and donor (Don) groups (Appendix A); propisochlor possessed a benzene ring, –C(=O)–, –Cl, Acc, Aro, and Don groups (Appendix A); metolachlor possessed a benzene ring, –C(=O)–, –Cl, –C–, Acc, Aro, hydrophobic (Hyd), and AtomQ groups (Appendix A); alachlor possessed a benzene ring, –C(=O)–, –Cl, –C–, –O–, Acc, Aro, Hyd, and AtomQ groups (Appendix A); acetochlor possessed a benzene ring, –Cl, –C–, –O–, Acc, Aro, Hyd, and AtomQ groups (Appendix A); pretilachlor possessed a benzene ring, –C(=O)–, –Cl, –C–, Acc, Aro, Hyd, and AtomQ groups (Appendix A); and butachlor possessed a benzene ring, –C(=O)–, –Cl, –C–, Acc, Aro, Hyd, and AtomQ groups (Appendix A).

The high-frequency residues (≥4) that participated in hTRPA1-AHs interactions were Asn722, Lys787, Gln791, Met844, Glu864, Lys868, and Leu982 (Appendix A). Furthermore, the cryo-EM structure of hTRPA1 identified a key interaction domain located in the transmembrane region and part of the extracellular region (Appendix A). Therefore, the high-frequency residues involved in hTRPA1-propanil included Asn722, Lys787, Gln791, and Leu982 (Figure 5A, Appendix A); those of hTRPA1-propisochlor included Lys787, Gln791, Met844, Lys868, and Leu982 (Figure 5B, Appendix A); those of hTRPA1-metolachlor included Lys787, Gln791, Met844, Glu864, Lys868, and Leu982 (Figure 5C, Appendix A); those of hTRPA1-alachlor included Asn722, Lys787, Gln791, Glu864, Lys868, and Leu982 (Figure 5D, Appendix A); those of hTRPA1-acetochlor included Asn722, Lys787, Gln791, Met844, Lys868, and Leu982 (Figure 5E, Appendix A); those of hTRPA1-pretilachlor included Asn722, Lys787, Met844, Glu864, Lys868, and Leu982 (Figure 5F, Appendix A); and those of hTRPA1-butachlor included Asn722, Lys787, Gln791, Glu864, Lys868, and Leu982 (Figure 5G, Appendix A).

The main types of interaction between hTRPA1-AHs included hydrophobic interactions and hydrogen bonding. The main types of interaction involved in hTRPA1-propanil included hydrophobic interaction of π-H bonding with Leu982 and hydrogen bonding with Asn722 (33.6%), Lys787 (32.7%), and Gln791 (22.9%); those in hTRPA1-propisochlor included π-H bonding with Leu982 and hydrogen bonding with Lys787 (63.7%), Gln791 (23.5%), Met844 (33.9%), and Lys868 (23.8%); those in hTRPA1-metolachlor included π-H bonding with Leu982 and hydrogen bonding with Lys787 (66.9%), Gln791 (65.3%), Met844 (33.2%), Glu864 (12.8%), and Lys868 (33.2%); those in hTRPA1-alachlor included π-H bonding with Asn722 and Leu982, as well as hydrogen bonding with Lys787 (22.8%), Gln791 (34.1%), Glu864 (24.8%), and Lys868 (27.8%); those in hTRPA1-acetochlor included π-H bonding with Leu982 and hydrogen bonding with Asn722 (23.8%), Lys787 (15.6%), Gln791 (16.9%), Met844 (53.2%), and Lys868 (22.0%); those in hTRPA1-pretilachlor included π-H bonding with Asn722 and Leu982, as well as hydrogen bonding with Lys787 (44.8%), Met844 (63.1%), Glu864 (26.8%), and Lys868 (54.8%); those in hTRPA1-butachlor included π-H bonding with Asn722 and Leu982, as well as hydrogen bonding with Lys787 (66.1%), Gln791 (23.9%), Glu864 (33.4%), and Lys868 (22.6%). For hTRPA1 interactions with propanil, propisochlor, metolachlor, alachlor, acetochlor, pretilachlor, and butachlor, the sum of the hydrogen bond scores was 89.2, 144.9, 211.4, 109.5, 131.5, 189.5, and 146.0, respectively (Appendix A); the interaction area was 247.60, 312.08, 277.85, 315.35, 324.98, 355.30, and 338.15 Å^2^, respectively (Appendix A); and the binding energy (BE) was −2.94, −2.71, −2.67, −2.64, −2.66, −2.56, and −2.55 kcal/mol, respectively (Appendix A). The interaction between AHs and Asn722 of hTRPA1 had two hydrogen bonds, that with Lys787 had seven, that with Gln791 had six, that with Met844 had four, that with Glu864 had four, and that with Lys868 had six. The interaction between AHs and Asn722 of hTRPA1 had three hydrophobic interactions, while that with Leu982 had seven. The order of hydrogen bonding between AHs and hTRPA1 from strongest to weakest was metolachlor > pretilachlor > butachlor > propisochlor > acetochlor > alachlor > propanil (Appendix A). The order of hTRPA1-AHs interaction area size was pretilachlor > butachlor > acetochlor > alachlor > propisochlor > metolachlor > propanil (Appendix A). The order of BE was propanil > propisochlor > metolachlor > acetochlor > alachlor > pretilachlor > butachlor (Appendix A). These results demonstrated that the molecular weight (MW) of AHs was inversely proportional to the absolute value of the binding energy (|BE|) (Appendix A). Therefore, a smaller MW AHs and larger |BE| represented a stronger hTRPA1-AHs interaction.

## 4. Discussion

Although lung cancer caused by AHs has been reported, there is paucity of reports on the toxic mechanisms of AHs poisoning and specific treatments [22]. Therefore, clarifying the mechanisms of AHs poisoning and exploring specific therapeutic drugs are urgently needed. This is the first study to suggest that the TRPA1 channel may provide a target for AHs poisoning and provides a new MTMC strategy for developing countermeasure drugs to treat inflammatory pathologies induced by AHs based on common mechanisms. Several toxic effects have been associated with pretilachlor, including endocrine disorder, cell apoptosis, and immunotoxicity in zebrafish embryos [37]. Research undertaken in an agricultural setting demonstrated that retinal degeneration increases with metolachlor and alachlor exposure and that risk of lung or pancreatic cancer is increased following exposure to acetochlor [24,38]. In addition, cytotoxicity, apoptosis, and expression of caspase-3 and caspase-9 in A549 cells are reportedly increased with acetochlor treatment [22]. However, the toxic target of AHs that induces inflammatory reactions remains undefined [22,24]. The present research evaluated the cytotoxicity of seven AHs on lung type II epithelial A549 cells in vitro for first time. The order of cytotoxicity induced by AHs in A549 cells from strongest to weakest was metolachlor > acetochlor > propisochlor > alachlor > butachlor > propanil > pretilachlor.

The concept of TRPs as chemosensors involved in detection and effectors of toxin action is intensively discussed in the cytotoxicity community at present. Previous studies have demonstrated that TRPA1 is a mediator of lung inflammation [8,39,40]. For example, TRPA1 was directly activated by acrolein or butenal inhalation in guinea pigs, whereas treatment with HC-030031 significantly relieved cough or reduced levels of sulfhydryl molecules such as acrolein, AITC, and cinnamaldehyde, as well as intracellular calcium concentrations and inflammatory reactions in mice [41,42]. Given these results, the current study explored whether TRPA1 is a common target for AHs poisoning in A549 cells. Seven AHs increased mRNA expression of TRPA1, whereas HC-030031 intervention or *TRPA1*-KO decreased TRPA1 mRNA expression and cytotoxicity. The results suggest that the TRPA1 channel could be a key common target for AHs poisoning. There are complicated cytotoxicity effects resulting from exposure to AHs, and we will explore the relevant oxidative stress injury and inflammatory reaction of A549 cells by AHs-treated via the TRPA1 channel or other receptors in the future.

Studies have shown that TRPA1-mediated calcium influx is the main mechanism of cytotoxicity caused by various chemicals, causing an intracellular cascade reaction leading to apoptosis by calcium influx [8,43]. The TRPA1 channel mediates lipopolysaccharide- or cigarette-smoke-induced calcium influx, cellular damage, and inflammatory reactions in vivo (such as in mice) and in vitro (such as in A549 cells), with increased expression of genes encoding caspase-1, IL-1β, IL-18, and other inflammatory factors. Meanwhile, inhibition or KO of TRPA1 can result in decreased calcium influx, inflammatory response, and cellular damage [44,45,46]. Collectively, these studies demonstrate that TRPA1-mediated calcium influx triggers cellular cascade events, including inflammation or apoptosis, in the lung epithelium. Meanwhile, in the current study, calcium influx was significantly increased in AHs-treated A549 cells in a dose-dependent manner, which was reduced by HC-030031 intervention or *TRPA1*-KO. Some studies have suggested that a calcium overload-mediated caspase-dependent apoptosis pathway via TRPA1 mediates apoptosis [8,22,24,39]. Thus, the TRPA1 channel may be a common target mediating calcium influx induced by AHs, while blocking or KO of this channel may have a protective effect (Figure 6).

Considering TRPA1 as a potential common target for AHs poisoning, exploring interactions between TRPA1 and different AHs was vital. X1-NH-X2 or Cl-C(H_2_)-X3 structures were present in hTRPA1-AHs displaying the highest binding affinity. LSPR analysis revealed the strongest affinity between propanil and TRPA1, whereas molecular docking simulations suggested this interaction could be due to exposure of the X1-NH-X2 nitrogen group. The strong affinity of TRPA1 for pretilachlor and metolachlor was likely related to the Cl-C(H_2_)-X3 structure. Common interaction mechanisms of hTRPA1-AHs involved hydrophobic (benzene ring, hydrocarbon chain) and active (–Cl, –C(=O)– or –O–, –N– or –NH–) groups, in which the benzene ring was key for interaction with TRPA1 [17,18]. The high-frequency residues involved in hTRPA1-AHs included Asn722, Lys787, Gln791, Met844, Glu864, Lys868, and Leu982, among which Lys787 and Leu982 were key residues. Common toxicity-effector groups of AHs were Acc, Aro, and Hyd. Common structures of AHs included the benzene ring (hydrophobic group), –C(=O)–, and –Cl (active group) groups, and hydrogen bonding and hydrophobic interactions were commonly involved in hTRPA1-AHs.

## 5. Conclusions

In conclusion, this study investigated common mechanisms underlying the cytotoxic effects induced by AHs in A549 cells, thus identifying potential therapeutic targets for the design of broad-spectrum antitoxins. This is the first study to systematically evaluate cytotoxicity and TRPA1 activation in A549 cells induced by AHs (propanil, propisochlor, metolachlor, alachlor, acetochlor, pretilachlor, and butachlor), revealing that the TRPA1 channel is a potential common target for AHs poisoning, inducing a series of intracellular cascades mediated by calcium influx that result in cytotoxic damage. Furthermore, the common molecular mechanism of AHs activating the TRPA1 channel was clarified, revealing common structures, active groups, key residues, hydrogen bonding, interaction areas, |BE|, and toxicity-effector groups. These findings will help contribute to the development of specific drug antagonists to weaken the cytotoxic effects of AHs. Meanwhile, our findings provide a scientific basis for a new MTMC strategy. Thus, the TRPA1 channel may provide a potential new target for the development of broad-spectrum antitoxins (especially against toxins with a benzene ring), which is increasingly important for ecological security and environmental safety.

## Figures and Tables

**Figure 1 ijerph-19-07985-f001:**
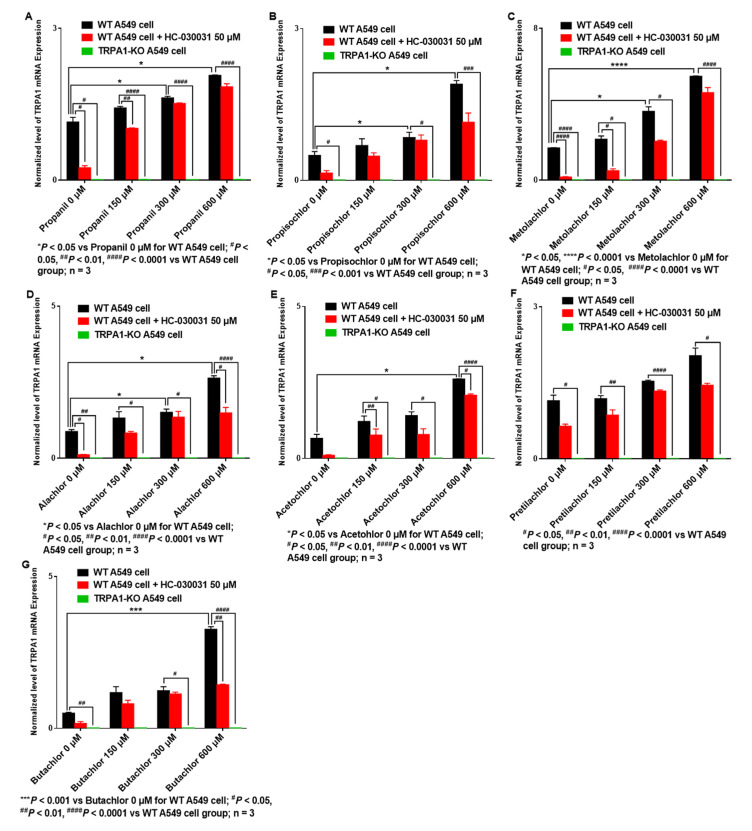
HC-030031 intervention or *TRPA1*-KO decreases expression of TRPA1 mRNA in A549 cells treated with AHs. (**A**) Propanil, (**B**) Propisochlor, (**C**) Metolachlor, (**D**) Alachlor, (**E**) Acetochlor, (**F**) Pretilachlor, (**G**) Butachlor.

**Figure 2 ijerph-19-07985-f002:**
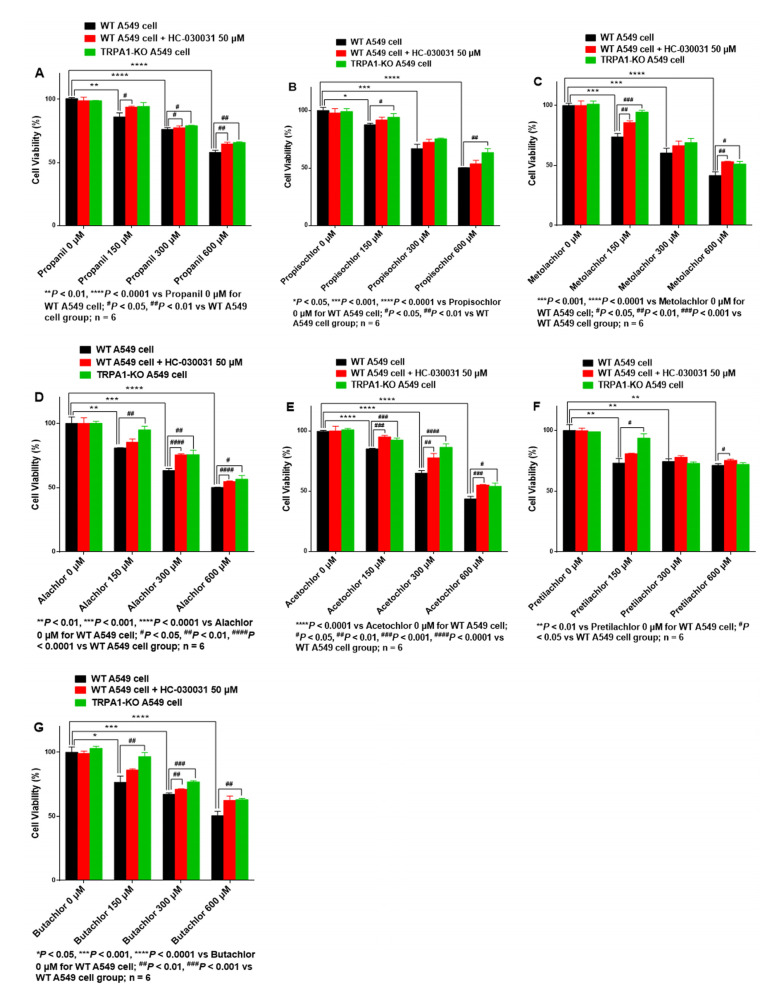
HC-030031 intervention or *TRPA1*-KO increases cell viability of AHs toward A549 cells. (**A**) Propanil, (**B**) Propisochlor, (**C**) Metolachlor, (**D**) Alachlor, (**E**) Acetochlor, (**F**) Pretilachlor, (**G**) Butachlor.

**Figure 3 ijerph-19-07985-f003:**
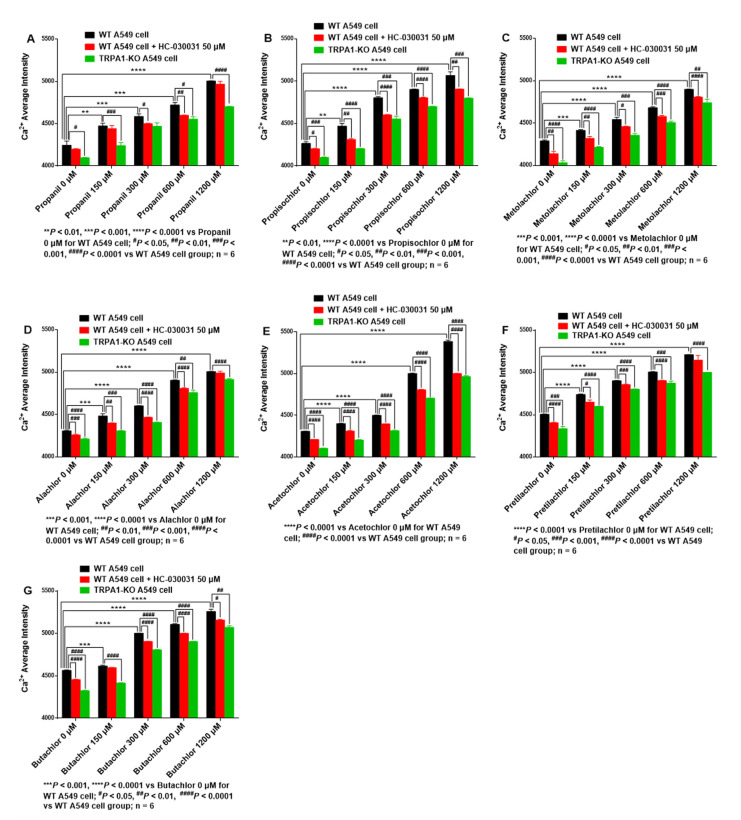
HC-030031 intervention or *TRPA1*-KO decreases calcium influx in A549 cells treated with AHs. (**A**) Propanil, (**B**) Propisochlor, (**C**) Metolachlor, (**D**) Alachlor, (**E**) Acetochlor, (**F**) Pretilachlor, (**G**) Butachlor.

**Figure 4 ijerph-19-07985-f004:**
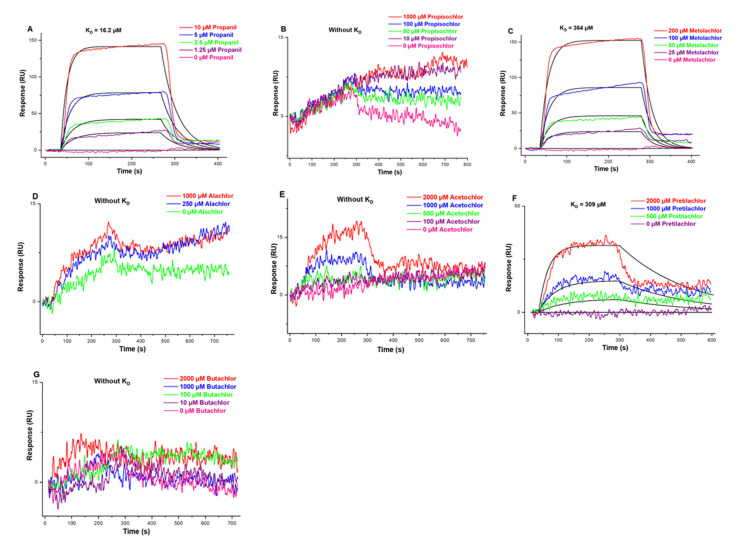
Affinity curves and constants of hTRPA1 and AHs. (**A**) Propanil, (**B**) Propisochlor, (**C**) Metolachlor, (**D**) Alachlor, (**E**) Acetochlor, (**F**) Pretilachlor, (**G**) Butachlor. K_D_: affinity constant.

**Figure 5 ijerph-19-07985-f005:**
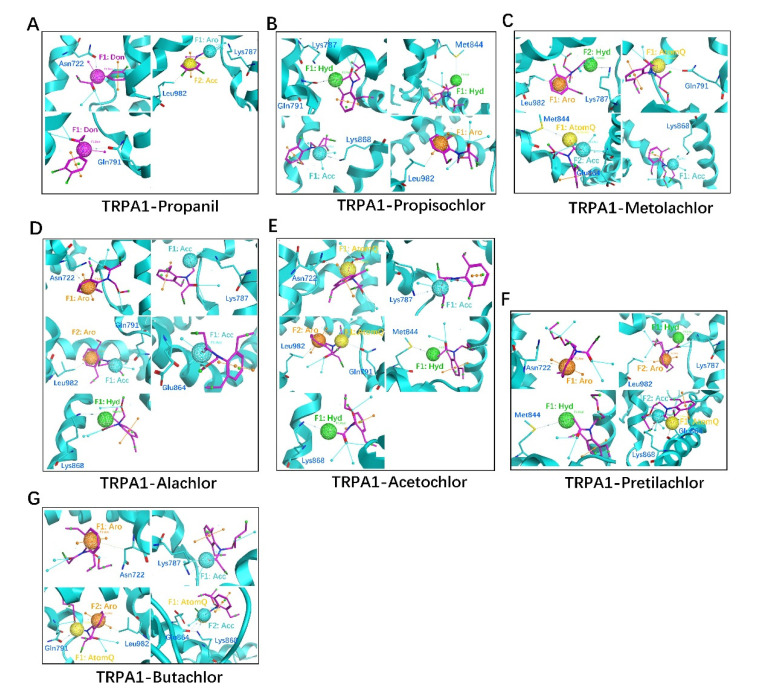
Interactions between hTRPA1 and AHs determined by molecular docking simulations. (**A**) Propanil, (**B**) Propisochlor, (**C**) Metolachlor, (**D**) Alachlor, (**E**) Acetochlor, (**F**) Pretilachlor, (**G**) Butachlor.

**Figure 6 ijerph-19-07985-f006:**
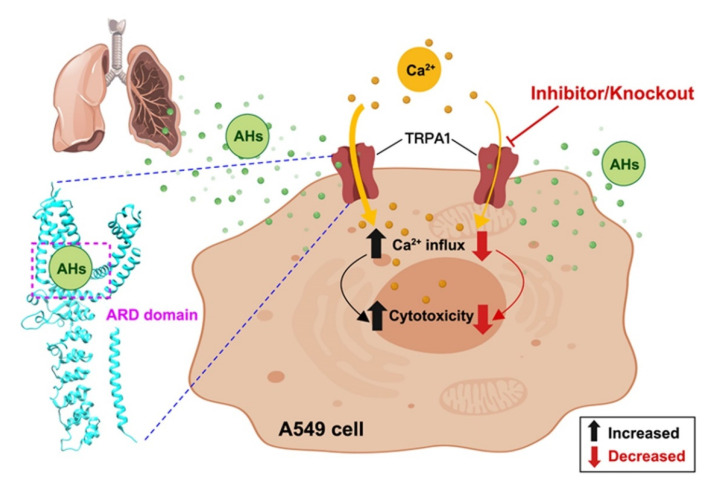
Cytotoxicity mechanism in A549 cells via TRPA1 channel by AHs poisoning.

**Table 1 ijerph-19-07985-t001:** IC_50_ values of AHs in WT/*TRPA1*-KO A549 cells.

AHs	A549 Cells (μM)	A549 Cells + HC-030031 50 μM (μM)	TRPA1-KO A549 Cells (μM)
Propanil	831	2290	2558
Propisochlor	564	1690	1828
Metolachlor	430	620	562
Alachlor	565	731	659
Acetochlor	524	647	696
Pretilachlor	2333	4456	3672
Butachlor	619	1757	767

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
