# Peer review of "Potential Common Mechanisms of Cytotoxicity Induced by Amide Herbicides via TRPA1 Channel Activation"

_ijerph, 2022, doi:10.3390/ijerph19137985_

Round 1

Reviewer 1 Report

In the present manuscript, the authors concluded that TRPA1 channel induction is responsible for the various AH compound's cytotoxic effects on A549 cells. Data seems to implicate that, but authors need more evidence to conclude that it is the TRAP1 that is playing a significant role in the Cytotoxicity. In the manuscript lot of important information is missing in the methods to understand the results. 

Specific comments are as follows:   

1.     Please specify how the real-time PCR data is analyzed and presented in the manuscript. Also, Y-axis labels are confusing, and the axis title does not reflect anything.

2.     For MS/MS analysis, how the authors precisely excised the TRAP1 protein band. Did they separate A549 cell lysate or the recombinant protein? The methods section should give precise details about how the experiments were done. Similarly, in LSPR, did the authors use recombinant protein or cell lysates?

3.     The results section is confusing with different numbers. Why are the same results presented in different sections? What is the difference between sections 3.2.1 vs. 3.3.2?  

4.     Since the treatment involves the different concentrations of the compounds and is all dissolved in DMSO, authors should include DMSO control.

5.     Author's subheading' 3.3.1. AHs treatment decreases TRPA1 mRNA expression in A549 cells, which is reversed by HC-030031 intervention or TRPA1-KO' is wrong because Ah treatments increased mRNA expression.

6.     The author needs to titrate different concentrations of the inhibitor HC030031 because the inhibitor has minimal influence on the cytotoxic effects at higher AH concentrations. Maybe increasing HC030031 concentration should have a more profound effect.

7.     It is understood that TRAP1 KO cells do not have any mRNA expression after AH treatment because the gene is knocked out. Looking at the overall data, TRAP1 KO alone does not bring back AH compounds' cytotoxic effects. 

Author Response

Reviewer #1

We thank Reviewer #1 for carefully appraising our work, offering insightful comments, and constructive suggestions to enhance our article. The manuscript has been extensively edited by Editage (www.editage.cn) to ensure that English language and style are free of errors. We have also adequately described the methods, clearly presented the results, and comprehensively expounded the conclusions to make sure the design of research is appropriate.

Point 1: Please specify how the real-time PCR data is analyzed and presented in the manuscript. Also, Y-axis labels are confusing, and the axis title does not reflect anything.

Response:

Thank you very much for the insightful comments and suggestions. Calculations were performed using the CFX ManagerTM Version 1.0 software (Bio-Rad Laboratories Ltd., USA) and are represented as fold-change in expression [2ΔΔC(t)] on a linear scale, which was specified in section 2.6 of the revised manuscript. The Y-axis titled “TRPA1 mRNA Expression” was changed to “Normalized level of TRPA1 mRNA Expression” in Figure 2, which was revised in section 3.2.3 of the manuscript.

Point 2: For MS/MS analysis, how the authors precisely excised the TRPA1 protein band. Did they separate A549 cell lysate or the recombinant protein? The methods section should give precise details about how the experiments were done. Similarly, in LSPR, did the authors use recombinant protein or cell lysates?

Response:

For MS/MS analysis, the TRPA1 protein band, which was the recombinant protein, was precisely excised from the gel kept on a clean glass plate using a sharp blade. The methods section 2.8. contains details about the protocol. The authors used TRPA1 recombinant protein in LSPR, which is included in section 2.9 of the revised manuscript.

Point 3: The results section is confusing with different numbers. Why are the same results presented in different sections? What is the difference between sections 3.2.1 vs. 3.3.2?

Response:

Initially, section 3.2.1. presented the results for AHs-treated A549 cell viability where viability decreased significantly, while section 3.3.2. presented the results for AHs-treated A549 cell viability also subjected to HC-030031 intervention or TRPA1-KO, where viability increased significantly. Therefore, the results presented in sections 3.2.1 and 3.3.2 are different. The revised supplementary material discusses the viability of A549 cells treated with AHs in figure S2 and the revised manuscript presented the viability of AHs-treated A549 cells increased with HC-030031 intervention or TRPA1-KO in figure 2, which were revised in revised manuscript of section 3.2.1 and 3.3.2, respectively.

Point 4: Since the treatment involves the different concentrations of the compounds and is all dissolved in DMSO, authors should include DMSO control.

Response:

Although the treatment involves different concentrations of compounds, DMSO at the concentration of 0.1% is safe for almost all cells. The concentrations of 0, 150, 300, 600, 1200 μM for AHs were obtained by diluting over one thousand times with 0.1% of DMSO; therefore, we have not included a DMSO control.

Point 5: Author's subheading' 3.3.1. AHs treatment decreases TRPA1 mRNA expression in A549 cells, which is reversed by HC-030031 intervention or TRPA1-KO' is wrong because AHs treatments increased mRNA expression.

Response:

Thank you for highlighting this mistake. We acknowledge this error and have corrected it. The subheading 3.3.1. was changed to “AHs treatment increased TRPA1 mRNA expression in A549 cells, which is reversed by HC-030031 intervention or TRPA1-KO”.

Point 6: The author needs to titrate different concentrations of the inhibitor HC030031 because the inhibitor has minimal influence on the cytotoxic effects at higher AHs concentrations. Maybe increasing HC030031 concentration should have a more profound effect.

Response:

In our preliminary experiment, we explored different concentrations of HC-030031 in A549 cells exposed to AHs (IC50) (not published). We found that 50 μM HC-030031 intervention could observably decrease the level of TRPA1 mRNA expression and calcium influx, increase the viability of AHs-treated A549 cells, and had the lowest toxicity. Moreover, 50 μM HC-030031 could attenuate TRPA1-mediated nociception and mechanical allodynia in models of inflammatory or neuropathic pain. Therefore, we affirmed that HC-030031 at a concentration of 50 μM is suitable to intervene AHs-treated A549 cytotoxicity.

Point 7: It is understood that TRPA1 KO cells do not have any mRNA expression after AHs treatment because the gene is knocked out. Looking at the overall data, TRPA1 KO alone does not bring back AHs compounds' cytotoxic effects.

Response:

The expression of TRPA1 mRNA was negligible in AHs-treated TRPA1-KO A549 cells treated, as determined by RT-qPCR tests. According to the CCK-8 assay, after AHs treatment, the viability of TRPA1-KO A549 cells was compared with the wide-type A549 cells.

References

  1. Nassini, R.; Fusi, C.; Materazzi, S.; Coppi, E.; Tuccinardi, T.; Marone, I. M.; De Logu, F.; Preti, D.; Tonello, R.; Chiarugi, A.; Patacchini, R.; Geppetti, P.; Benemei, S. The TRPA1 channel mediates the analgesic action of dipyrone and pyrazolone derivatives. J. Pharmacol. 2015, 172(13), 3397-3411.

Reviewer 2 Report

The manuscript described a study on the molecular mechanism of cytotoxicity induced by amide herbicides via TRPA1 channel activation. The study also proposed the development of TRPA1 as a promising therapeutic target for broad-spectrum antitoxins. The reviewer recommends publication of the manuscript after the authors can address the following concerns:

1) the concentration applied in the in vitro study in the manuscript is too high (e.g. 430, 524, 564, 565, 619, 831, and 2333 μM). The author should provide rationale for using such high concentrations (e.g. the actual in vivo concentrations of the amide herbicides in poisoning incidents).

2) the study seemed to only use A549 cell line for detailed molecular mechanisms. Usually, such in vitro studies should use two or more cell lines for comparison studies and validate the results of proposed molecular mechanisms .

3) since A549 is a lung cancer cell line, should primary human lung-airway cells be used for such toxicity studies to simulate compound poisoning process instead of using lung cancer cell lines? 

Author Response

Reviewer #2

We thank Reviewer #2 for carefully inspecting our work, offering insightful comments, thoughtful questions, and constructive suggestions to enhance our manuscript. The manuscript has been extensively edited by Editage (www.editage.cn) to ensure that English language and style are free of errors. We have also comprehensively expounded the conclusions to make sure the design of this research is appropriate.

Point 1: The concentration applied in the in vitro study in the manuscript is too high (e.g. 430, 524, 564, 565, 619, 831, and 2333 μM). The author should provide rationale for using such high concentrations (e.g. the actual in vivo concentrations of the amide herbicides in poisoning incidents).

Response:

Thank you very much for your kind comments. AHs such as alachlor or metolachlor can induce hypotoxicity, thus in animals or cells by toxicological experiments [1]. The concentrations used in our study are acceptable and have been decided based on relevant literature.

Reference:

  1. Kirrane, E. F.; Hoppin, J. A.; Kamel, F.; Umbach, D. M.; Boyes, W. K.; DeRoos, A. J.; Alavanja, M.; Sandler, D. P. Retinal degeneration and other eye disorders in wives of farmer pesticide applicators enrolled in the agricultural health study. J. Epidemiol. 2005, 161(11), 1020-1029.

Point 2: The study seemed to only use A549 cell line for detailed molecular mechanisms. Usually, such in vitro studies should use two or more cell lines for comparison studies and validate the results of proposed molecular mechanisms.

Response:

TRPA1 is highly expressed in human lung epithelial cells, and the lungs as the key organs to undergo damage on pesticides exposure. Moreover, A549 cells form confluent monolayers with Type II pulmonary epithelial cells characteristic morphology. It is common and appropriate to explore the common injury mechanisms in the lungs due to AHs treatment using A549 cells. We appreciate your suggestion, and in the future, we will verify our results in human MRC-5 fibroblasts and investigate the functional differences of TRPA1 in spatially specific expression. Furthermore, this study aimed to unveil the molecular mechanisms underlying the cytotoxicity of AHs and did not prioritize assessing its effects in different cell lines. Therefore, we did not choose multiple cell lines for comparison.

Point 3: Since A549 is a lung cancer cell line, should primary human lung-airway cells be used for such toxicity studies to simulate compound poisoning process instead of using lung cancer cell lines?

Response:

Primary human lung-airway cells could be used, however, they are difficult to obtain and culture. Therefore, A549 cells, which used widely in vitro observation for lung poisoning [1,2], we used instead.

The relevant reference of response for point 3 was as following:

  1. Foldbjerg, R.; Autrup, D. H. Cytotoxicity and genotoxicity of silver nanoparticles in the human lung cancer cell line, A549. Toxicol. 2011, 85(7), 743-750.
  2. Mazloum-Ravasan, S.; Madadi, E.; Mohammadi, A.; Mansoori, B.; Amini, M.; Mokhtarzadeh, A.; Baradaran, B.; Darvishi, F. Yarrowia lipolytica L-asparaginase inhibits the growth and migration of lung (A549) and breast (MCF7) cancer cells. J. Biol. Macromol. 2021, 170(9), 406-414.

Reviewer 3 Report

The authors of the manuscript titled "Potential common mechanisms of cytotoxicity induced by amide herbicides via TRPA1 channel activation" report the cytotoxicity, and expression of transient receptor potential cation channel subfamily A member 1 (TRPA1) mRNA, and intracellular calcium influx in A549 cells exposed to amide herbicides. The body of work presented here is appropriate for Molecules however, it needs some minor revisions before it can be considered for publication.

Points that needed to be addressed.

1.      The authors claim that the TRPA1 channel could be a key common target for AHs poisoning. However, there was a decrease in cell viability even in TRPA1-KO A549 cells which suggests an additional mechanism responsible for the cytotoxicity effects. Did the authors explore this possibility and the overall cytotoxicity effects of AHs towards A549 could be because of a combination of factors and not just because of TRPA1 channel? Please clarify that in the discussion.

2.   In general, the authors need to improve the quality of the figures in their manuscript. Substituting the compounds with numbers and enlarging the panel with quality along with better legible axes and contents could significantly improve the quality of this manuscript.

3.    Do the authors have any input on why the WT A549 and TRPA1-KO cells have similar cell viability at concentrations 300 mM and 600 mM (Figure 1, panel F).

4.   Please enlarge the panels of figure 4 so that the axes and content of the graphs are more legible to the readers.

5.       The authors should increase the clarity of the figure 5. Resize the panels so that the structures and amino acids residues are legible.

6.       The panel A of figure S5 is too small to read. Please enhance the panel with clarity.

Author Response

Reviewer #3

We thank Reviewer #3 for carefully appraising our work, offering insightful comments, and constructive suggestions to improve our article. The manuscript has been extensively edited by Editage (www.editage.cn) to ensure that English language and style are free of errors. We have also sufficiently provided background information, added relevant references, clearly presented the results, and comprehensively expounded the conclusions to make sure the design of research is appropriate.

Point 1: The authors claim that the TRPA1 channel could be a key common target for AHs poisoning. However, there was a decrease in cell viability even in TRPA1-KO A549 cells which suggests an additional mechanism responsible for the cytotoxicity effects. Did the authors explore this possibility and the overall cytotoxicity effects of AHs towards A549 could be because of a combination of factors and not just because of TRPA1 channel? Please clarify that in the discussion.

Response:

Thank you very much for the insightful comments and suggestions. TRPA1 channel is a key common target for AHs poisoning. The overall cytotoxicity may be complex and involving several factors, which has been discussed in line 503-507 We will explore this possibility and the overall cytotoxicity o in the future.

Point 2: In general, the authors need to improve the quality of the figures in their manuscript. Substituting the compounds with numbers and enlarging the panel with quality along with better legible axes and contents could significantly improve the quality of this manuscript.

Response:

 We have improved the quality of the figures in the manuscript and enlarged the panel along with better legible axes and contents to.

Point 3: Do the authors have any input on why the WT A549 and TRPA1-KO cells have similar cell viability at concentrations 300 mM and 600 mM (Figure 1, panel F).

Response:

We aim to clarify the reasons for the similar cell viability at 300 μM and 600 μM concentrations pretilachlor in WT/TRPA1-KO A549 cells in future studies. We speculate that this may be related to the structure of pretilachlor, the sensitivity of A549 cells to pretilachlor or the inhibitory effect of A549 cells by higher concentration of pretilachlor.

Point 4: Please enlarge the panels of figure 4 so that the axes and content of the graphs are more legible to the readers.

Response:

Thanks you. We have enlarged the panels inn figure 4 for improving legibility.

Point 5: The authors should increase the clarity of the figure 5. Resize the panels so that the structures and amino acids residues are legible.

Response:

Thank you. We have increased the clarity of figure 5 and resized the panels so that the structures and amino acids residues are legible.

Point 6: The panel A of figure S5 is too small to read. Please enhance the panel with clarity.

Response:

Thank you. We have enhanced panel A of figure S5.

Round 2

Reviewer 1 Report

The authors have adequately answered all my questions and so accepted the article for publication.

This manuscript is a resubmission of an earlier submission. The following is a list of the peer review reports and author responses from that submission.